# Ethylene Inhibition Reduces De Novo Shoot Organogenesis and Subsequent Plant Development from Leaf Explants of *Solanum betaceum* Cav.

**DOI:** 10.3390/plants12091854

**Published:** 2023-04-30

**Authors:** Mariana Neves, Sandra Correia, Jorge Canhoto

**Affiliations:** 1Centre for Functional Ecology, TERRA Associate Laboratory, Department of Life Sciences, University of Coimbra, 3000-456 Coimbra, Portugal; mariananevespt@gmail.com (M.N.); sandraimc@ci.uc.pt (S.C.); 2InnovPlantProtect CoLab, 7350-478 Elvas, Portugal

**Keywords:** ACC, aminoethoxyvinylglycine, ethylene modulation, ethephon, in vitro culture, organogenesis, silver nitrate, tree tomato

## Abstract

In de novo shoot organogenesis (DNSO) plant cells develop into new shoots, without the need of an existing meristem. Generally, this process is triggered by wounding and specific growth regulators, such as auxins and cytokinins. Despite the potential significance of the plant hormone ethylene in DNSO, its effect in regeneration processes of woody species has not been thoroughly investigated. To address this gap, *Solanum betaceum* Cav. was used as an experimental model to explore the role of this hormone on DNSO and potentially extend the findings to other woody species. In this work it was shown that ethylene positively regulates DNSO from tamarillo leaf explants. Ethylene precursors ACC and ethephon stimulated shoot regeneration by increasing the number of buds and shoots regenerated. In contrast, the inhibition of ethylene biosynthesis or perception by AVG and AgNO_3_ decreased shoot regeneration. Organogenic callus induced in the presence of ethylene precursors showed an upregulated expression of the auxin efflux carrier gene *PIN1*, suggesting that ethylene may enhance shoot regeneration by affecting auxin distribution prior to shoot development. Additionally, it was found that the de novo shoot meristems induced in explants in which ethylene biosynthesis and perception was suppressed were unable to further develop into elongated shoots. Overall, these results imply that altering ethylene levels and perception could enhance shoot regeneration efficiency in tamarillo. Moreover, we offer insights into the possible molecular mechanisms involved in ethylene-induced shoot regeneration.

## 1. Introduction

Ethylene is a gaseous plant hormone involved in several physiological processes, including plant growth and development, fruit ripening and seed germination [1]. Its effect on plant development encompasses inhibition on primary root growth [2] and lateral root formation [3] and a positive modulation of root hair formation and growth [4,5]. Ethylene is also involved in the inhibition of leaf growth due to its regulation of cell division and cell expansion (reviewed in [1]). Within the most varied roles of ethylene, its involvement in plant stress responses is notable. Ethylene is a key regulator of stress adaptation, mediating both abiotic [6] and biotic [7] stresses. This hormone acts as a signaling molecule, inducing a variety of physiological and biochemical changes and enabling plants to cope with environmental stress [8].

Plant regeneration systems, such as de novo shoot organogenesis (DNSO) or somatic embryogenesis, rely on plant cell plasticity, which is inherent to mechanisms of pluripotency/totipotency. It can be induced or enhanced by exogenous stress stimulus, such as wounding and hormonal treatments [9,10]. These plant regeneration processes are divided in different morphological stages, often in response to a balance between auxins and cytokinins [10]. DNSO is generally divided in pluripotency acquisition (cell dedifferentiation), shoot promeristem formation and shoot development [11].

DNSO has been extensively studied in *Arabidopsis thaliana*, typically involving an initial culture step in an auxin-rich medium (pluripotency acquisition) and posterior subculture in a cytokinin-rich medium (promeristem formation and shoot development) [11,12]. Generally, pluripotency acquisition occurs not only in response to exogenous auxin supplementation but is also triggered by wounding [10]. Wounding positively modulates callus formation with a marked accumulation of cytokinin at cutting sites and contributes to further organ regeneration [12,13]. In addition, ethylene biosynthesis is also triggered by wounding [14] and in response to cytokinin [15,16] and auxin [17] treatments. There is compelling evidence that callus formation and the subsequent ability to regenerate is derived from specific cell types that act as potential pluripotent stem cells [18,19]. Auxin-induced callus appears to originate from pericycle and pericycle-like cells located around the vasculature, while wound-induced callus can arise from various cell types such as epidermis, xylem parenchyma, procambium, and mesophyll [18]. In some species, de novo shoots and roots regenerate from procambium or cambium cells [18,19]. 

The effect of ethylene modulation on in vitro regeneration, focusing on DNSO and somatic embryogenesis was recently reviewed [20]. Ethylene seems to affect in vitro culture depending on the species or the explants used. Some studies point out a negative effect of ethylene on DNSO in *Cucumis melo* [21,22] and *Brassica juncea* [23,24]. Nevertheless, in *Solanum pennellii* [25] and *Arabidopsis thaliana* [26] ethylene perception seems to be required. For somatic embryogenesis, the role of ethylene in reverting recalcitrance in genotypes with low regeneration capacity was already described [27]. 

Ethylene can act in plant regeneration as a stress responsive agent in addition to its hormonal effect. The APETALA2/Ethylene responsive factor (AP2/ERF) transcription factors family has been highlighted for their regulation in multiple stress responses [28]. Some of these transcription factors respond to ethylene and promote the activation of ethylene-dependent responsive genes [28,29]. Regarding plant regeneration, in *Medicago truncatula,* a transcription factor of this family induced by ethylene, designated MtSERF1, seems to be required for somatic embryo development in the presence of auxin and cytokinin [30]. Likewise, in *A. thaliana* and *Glycine max*, orthologs of this transcription factor were also described with a positive correlation in somatic embryo development in the presence of auxin. 

*Solanum betaceum* Cav., commonly known as tamarillo, is an Andean solanaceous tree, that has been used as a model system to study several micropropagation/regeneration processes, such as organogenesis [31,32,33,34] and somatic embryogenesis [35,36,37]. It has allowed a better understanding of these systems, and the possibility of further applications in other species to optimize protocols and regenerate adult selected trees [38]. 

In tamarillo, the effect of plant growth regulators, such as auxins and/or cytokinins on DNSO was already tested [31,32,33,34]. For instance, thidiazuron (TDZ) [31,32], benzylaminopurine (BAP) [31,33,34] or combinations of BAP and naphthaleneacetic acid [33,34] demonstrated to be the most suitable inducers of this process. In terms of initial explant, leaves proved to be the most effective explant for the induction of shoot regeneration, when compared to petioles or root seedlings [31,33]. 

DNSO has been highly applied for breeding purposes, especially in dicotyledonous species, due to the simplicity and robustness of culture conditions [9]. Ethylene response varies significantly based on the organ, time, and species; thus it is difficult to assign a unique and general role of ethylene in the regulation of biotechnological processes [39]. Therefore, understanding how ethylene regulates DNSO, focusing on woody species, could have important practical applications, such as developing new methods for plant propagation, regeneration and giving relevant insights about recalcitrance in some species.

The aim of this work was to evaluate the effect of ethylene on DNSO, from leaf explants of tamarillo. To achieve this goal, leaf explants were cultured in the presence of different ethylene modulators and their effects evaluated on shoot regeneration. The impact of this modulation on subsequent plant development from the regenerated shoots was also assessed. Finally, the expression of genes related to ethylene biosynthesis, in particular *ACS1* and *ACO1*, of the transcription factor *ERF061* and of the auxin efflux carrier *PIN1*, were evaluated to unveil possible molecular mechanisms behind the ethylene modulation effect. 

## 2. Results

### 2.1. Effect of Ethylene Modulation on De Novo Shoot Organogenesis

To test the effect of ethylene on DNSO in tamarillo, leaf explants were cultured in the presence of 10 µM of each different ethylene modulator (Figure 1a). Silver nitrate (AgNO_3_) was used to inhibit ethylene perception and aminoethoxyvinylglycine (AVG) to inhibit ethylene biosynthesis. To stimulate ethylene action on plant tissues, the ethylene precursors, 1-aminocyclopropane-1-carboxylic acid (ACC) and 2-chloroethylphosphonic acid (commercially known as ethephon; ETH) were applied. The protocol for shoot regeneration involves 3 weeks in the dark followed by 5 weeks in a 16 h photoperiod (Figure 1b). These different ethylene modulators were present in culture medium during all the induction process. At the end of 8 weeks, the effect of each modulator on shoot regeneration percentage, the number of buds and shoots developed per explant and the morphology of the regenerated shoots (Figure 1c) were evaluated. 

Callus formation was observed at the end of the third week at wounding sites (Figure 1d). Leaf explants exposed to inhibitors of ethylene biosynthesis or perception presented a reduced capacity for callus induction (Figure 1d(ii,iii)). Further shoot development and elongation was also compromised in ethylene inhibition treatments (Figure 1c(xi)). Furthermore, some abnormal leaf shape in shoots regenerated in the presence of AgNO_3_ (Figure 1c(iv)) were noticed. Few shoots regenerated in ACC treatments presented signs of hyperhydricity (Figure 1d(iv—left)), which was completely reversed when shoots were subcultured in hormone-free MS medium. 

Regeneration percentage was not significantly affected by any treatment (Table 1). However, this parameter seemed to decrease across all conditions, with a marked reduction in AVG treatment (around 48%). The number of buds and shoots regenerated per responsive explant was the parameter significantly affected by ethylene modulation. We found a statistically significant increase in the number of buds and shoots developed when ethylene perception was enhanced by ACC and ETH treatments. The inhibition of ethylene perception or biosynthesis by AgNO_3_ and AVG significantly reduced the number of the buds and shoots regenerated per explant. 

AgNO_3_ and AVG treatments decreased almost two-fold the number of buds and shoots regenerated per explant (around six/explant) relative to control conditions (around 12/explant). When the enhancement of ethylene availability by ACC and ETH was compared with the inhibition of its perception and biosynthesis by AgNO_3_ and AVG, it was found a three-fold increase in the number of buds and shoots regenerated per explant when ethylene perception is enhanced (19/explant for ACC and 18/explant for ETH) and vice-versa. Interestingly, opposite effects on ethylene modulation reduced or increased the number of buds or shoots per explant by around six explants compared to the control condition. Moreover, similar modulation treatments contributed to similar effects on regeneration.

### 2.2. Effect of Ethylene Modulation on the Expression of Ethylene-Related Genes and Auxin Efflux Carrier 

The effect of ethylene modulation on the expression of specific genes was evaluated in two culture timepoints. Samples from the third week of culture, right before the transition to light and at the end of 8 weeks, defined as the final point of regeneration protocol (Figure 2a), were collected and analyzed. 

Regarding ethylene biosynthesis, the expression of *ACS1* and *ACO1* (Figure 2b–c) was assessed. These genes encode for two specific isoforms of the main enzymes involved in ethylene biosynthesis, ACC synthase and ACC oxidase, respectively. In both timepoints, we found a statistically significant increase in *ACS1* expression for the ETH condition (2.93 and 1.88 log2 fold). At the end of the eighth week, AVG promoted the opposite effect, with a statistically significant decrease of *ACS1* expression (−0.85 log2 fold). AgNO_3_ and ACC treatments also showed a tendency to upregulate *ACS* expression at both timepoints. 

For *ACO1* expression (Figure 2c), a statistically significant downregulation in AgNO_3_ treatments for the first time point (−0.39 log2 fold) was found. At the end of 8 weeks, *ACO1* expression tends to be upregulated in all conditions.

The expression of the gene *ERF061*, encoding for a transcription factor of the AP2/ERF superfamily was also evaluated. *ERF061* was significantly upregulated in the presence of the ethylene precursor ETH, but also in the presence of the ethylene biosynthesis inhibitor AVG, post 3 weeks of culture (Figure 2d). However, the upregulation of this gene was more notorious in ETH treatments (6.36 log2 fold) than in AVG treatments (1.95 log2 fold). *ERF061* expression remained significantly upregulated in the presence of ETH post 8 weeks (3.72 log2 fold). Similarly, we observed a statistically significant upregulation of this gene in ACC presence (1.92 log2 fold) for the same culture timepoint. The upregulation observed previously in AVG treatment for the first timepoint did not remain at the end of the eighth week, in contrast to ETH treatment. Moreover, in this second timepoint, although not statistically significant, *ERF061* expression seemed to be downregulated when ethylene perception and biosynthesis were inhibited. 

Finally, the effect of ethylene modulation in the expression of the auxin efflux carrier *PIN1* (Figure 2e) was assessed. Before transition to light, a statistically significant increase in *PIN1* expression was found in the presence of ethylene precursors ACC (1.24 log2 fold) and ETH (1.06 log2 fold). Furthermore, *PIN1* expression was significantly downregulated in the presence of the ethylene biosynthesis inhibitor AVG (−1.72 log2 fold). *PIN1* showed a tendency to be also downregulated when ethylene perception is inhibited by AgNO_3_. No significant statistical differences were found at the end of the eighth week. Nevertheless, *PIN1* expression values showed a tendency to remain upregulated or downregulated in the ACC and AVG conditions, respectively.

### 2.3. In Vitro Rotting, Plant Development and Acclimatization

After 8 weeks, the 1 cm shoots regenerated in each condition were cultured in hormone-free MS medium for 1 month to induce rooting and shoot development (Figure 3a). Occasionally, shoots can be cultured in MS medium with low BAP concentrations to induce shoot development and elongation, before the rooting induction step. Nevertheless, this step was skipped to avoid additional hormonal stimuli that could mask ethylene modulation in the subsequent development of the regenerated shoots.

After 1 month in hormone-free MS medium, all shoots regenerated in control conditions and in the presence of ethylene precursors were successfully rooted with well-developed and elongated roots (Figure 3b,c). Interestingly, shoots regenerated in treatments in which ethylene perception and biosynthesis were inhibited had their capacity to develop adventitious roots disrupted (Figure 3b,c). Likewise, shoot development and elongation was also negatively affected. No significant differences in morphological parameters such as plant height, root length and number of roots per shoots, between control and both ACC and ETH treatments were found (Figure 3c).

Acclimatization was successfully achieved for plants regenerated from control and both ACC and ETH treatments (Figure 3d), reaching 83%, 89%, and 100%, respectively. No differences were found in plant height between treatments after 1- and 3-months of ex vitro growth (Figure 3d–f). Nevertheless, the plants developed from shoots regenerated in the presence of ACC presented a significantly higher dry matter percentage when compared to the control plants (Figure 3d) after 3 months in ex vitro conditions.

## 3. Discussion

### 3.1. Ethylene Positively Modulates Shoot Regeneration from Callus Induction to Shoot Development

Our results bring out a positive effect of ethylene on DNSO from tamarillo leaves, especially notorious in the number of buds and shoots regenerated per explant. While ethylene enhancement contributed to an effective increase in the number of regenerated shoots, the inhibition of its perception and biosynthesis had the opposite effect. A positive effect of ethylene was already described on DNSO from leaf explants of other Solanaceae, such as *Solanum pennellii* [25] and *Petunia hybrida* L. [40]. ACC and AgNO_3_ treatments affected *S. pennellii* regeneration in a similar manner as reported in our study. Likewise, exogenous ethylene applications or AgNO_3_ treatments increased or reduced the number of shoots per explant in *P. hybrida*, respectively. Ethylene is also essential to induce shoot regeneration from cotyledons explants in *Arabidopsis* [26].

Ethylene perception seems to be required to enhance pluripotent callus formation at the cutting sites of tamarillo leaf explants. In fact, less callus formation found in AgNO_3_ and AVG treatments negatively impacted further regeneration. Knowing pluripotent callus is formed at wounding sites in leaf explants, which later lead to shoot development [41], a positive correlation between ethylene perception, pluripotent callus formation, and the explant ability to regenerate can be assumed. 

Our protocol for DNSO from tamarillo leaves only englobes an exogenous source of cytokinin (BAP, 8.8 µM) and wounding as stress stimulus, although it is sufficient to promote callus formation and subsequent shoot regeneration. Callus development relies on cytokinin accumulation at cutting sites [13] and also in exogenous auxin supplementation [42]. Organ regeneration requires pluripotent acquisition in the middle of the cell layer of the callus promoted by auxin production and enhancement of cytokinin sensitivity [43]. It is known wounding and cytokinin enhance ethylene biosynthesis [14,15,16,44,45,46,47,48] and, in turn, ethylene also increases cytokinin levels [49]. Likewise, ethylene also enhances auxin biosynthesis and vice-versa [2,17,50,51]. In tomato leaves, wounding stimulates ethylene production [44] and the cytokinin BAP upregulates the *ACO-like* gene [52]. TDZ and BAP treatments also increase ethylene production in cotton leaves [45]. In our study, BAP treatment (in the control condition) is effective to achieve shoot regeneration from tamarillo leaf explants. Furthermore, other studies have already demonstrated TDZ and BAP as good plant growth regulators to promote DNSO from leaf explants in tamarillo [31,32,33,34]. Recently, Shin et al. [53] demonstrated that ethylene facilitates cell dedifferentiation and auxin-induced callus formation by regulating the abundance of transcripts for auxin receptor genes. This background supports our results and suggests that ethylene can positively affect regeneration in a crosstalk between cytokinin and auxins. Further research is needed to confirm this assumption, but we can hypothesize that, in control conditions, cytokinin and wounding stimulates ethylene production, which in turn enhance auxin biosynthesis contributing to pluripotent callus formation, without the requirement of exogenous auxin supplementation. In the presence of AgNO_3_ and AVG treatments the effect of cytokinin and wounding on ethylene production is reduced and consequently auxin biosynthesis is downregulated leading to a decrease in callus formation and subsequent shoot regeneration. In addition, this regulation can be addictively regulated by the presence of the ethylene precursors justifying the enhancement of shoot regeneration.

### 3.2. Ethylene Modulation Differentially Regulates Gene Expression Related to Ethylene Biosynthesis and ERF061 Depending on Regeneration Stage

We explored the expression of two genes encoding for ethylene biosynthetic enzymes, ACC synthase and ACC oxidase (*ACS1* and *ACO1*), and the transcription factor *ERF061* that potentially affects regeneration and could be regulated by ethylene. We based our decision to analyze *ERF061* expression on the knowledge that it has been considered a putative candidate gene related to shoot regeneration in *Arabidopsis* [54]. *ERF061* belongs to the same AP2/ERF subfamily of *WIND1*, which promotes callus formation and shoot regeneration in *Arabidopsis* [55]. The acquisition of regeneration competency is heavily dependent on the role of WIND1, and its ectopic expression increases de novo shoot regeneration from *Arabidopsis* root explants, without the need for either wounding or auxin pre-treatment [56].

An upregulation of *ACS1* expression by ETH treatment was found. In addition, ACC and AgNO_3_ also showed a tendency to upregulate its expression. On the contrary, a downregulation of *ACS1* was found in AVG treatments post 8 weeks. AVG is a strong inhibitor of ACC synthase activity [57,58] while AgNO_3_ inhibits ethylene action at the receptor level [59]. Besides the negative effect on ethylene action by both modulators, these compounds seem to affect *ACS1* gene expression differently. In agreement with our results, several *ACS* genes are downregulated by AVG in *Cucurbita maxima* [60] and *Pyrus bretschneideri* [61] while AgNO_3_ seems to only downregulate some *ACS* genes [60]. Both ethylene precursors and exogenous ethylene treatments also seem to upregulate some *ACS* genes [60,62,63,64], supporting our observations. In turn, the effect of ethylene modulation on *ACO1* expression was less notorious compared to those of *ACS1*. Only AgNO_3_ treatment significantly downregulated *ACO1* expression post 3 weeks of culture.

*ERF061* seems to be involved in biotic and abiotic stress responses [65,66]. Some studies also point that *ERF061* is upregulated by exogenous ethylene and ethephon treatments [67,68,69]. Effectively, we found in both ACC and ETH treatments an upregulation of *ERF061* expression at the eighth week. Contradictory results were observed at the third week, in which *ERF061* expression was not only upregulated by ETH, but also by the ethylene biosynthesis inhibitor AVG, albeit to a lesser extent. Nevertheless, at the end of the eighth week, the AVG effect shows the opposite trend while its upregulation by ETH is preserved. Recently, *ERF061* was postulated to be a possible transcription factor involved in regulating plant flower development and flower tissue formation in *Actinidia eriantha* [70]. Our results do not allow us to affirm *ERF061* expression is induced by ethylene. However, *ERF061* upregulation by ACC and ETH found at the end of 8 weeks, in which shoot-buds are being developed raises some questions about the possible involvement of *ERF061* in shoot regeneration, posterior to the callus formation stage. 

### 3.3. Ethylene Precursors Upregulate PIN1 Expression in Cytokinin-Induced Callus

PIN1 is an auxin efflux carrier required for an efficient shoot meristem induction in cytokinin-rich medium [71]. Interestingly, ACC upregulates *PIN1* expression in Arabidopsis roots [2]. Thus, we considered that a possible modulation of its expression by ethylene could impact shoot regeneration. In fact, one of the outstanding observations in our results was the significant upregulation of *PIN1* at the end of the third week in the presence of ethylene precursors ACC and ETH while its expression was significantly downregulated by AVG treatment. The presence of AgNO_3_ tends to also downregulate *PIN1* in the same culture timepoint_,_ although not significantly. In shoot regeneration from *Arabidopsis* root explants, *PIN1* is locally upregulated marking future sites of primordium initiation [71] and further development of shoot promeristem also requires its upregulation [9]. Moreover, *PIN1* loss of function mutation reduces shoot regeneration [71]. Therefore, we can assume the downregulation of *PIN1* in AVG and AgNO_3_ treatments explain their negative impact on shoot regeneration and postulate the requirement of ethylene for DNSO in tamarillo leaf explants.

### 3.4. Inhibition of Ethylene Biosynthesis and Perception during Shoot Regeneration Negatively Impacts Subsequent Plant Development

Ethylene also seems to be essential to plant development in tamarillo as the shoots regenerated in treatments where ethylene perception or biosynthesis was inhibited were visibly less developed. Subsequent development, such as shoot elongation and adventitious root (AR) formation was also disrupted. In *S. pennellii*, shoots regenerated in AgNO_3_ treatments were also less developed [25]. In agreement with our observations, in the woody species *Populus tremula*, AVG treatments also inhibited shoot elongation, induction and development of buds and root formation; in turn, ACC and ETH treatments stimulated these parameters [72]. The positive effect of ethylene in shoot proliferation of *P. tremula* raises the prospect of micropropagation protocols based on the action of ethylene produced by the plant itself instead of exogenous hormone treatment, such as the use of small-volume vessels with gas exchange restriction [73].

For biotechnological purposes subsequent AR formation is fundamental to a successful acclimatization. Studies have shown that ethylene can affect plant development differently, depending on the plant species, tissue type, and hormone supplementation [39,74]. In tomato and cucumber, ethylene also increases AR formation through an auxin-ethylene crosstalk [75,76,77], supporting our results. The inhibition of AR induction in shoots regenerated from AgNO_3_ and AVG impacted further acclimatization and ex vitro adaptation. Unknown molecular patterns previously induced in regenerated shoots seems to impact further AR initiation which is not reverted in the hormone-free MS medium. 

## 4. Materials and Methods

### 4.1. Plant Material

Leaves of red tamarillo shoots previously established from in vitro germinated seeds were used for shoot regeneration assays. Tamarillo shoots were in vitro propagated in MS medium [78] supplemented with sucrose (0.07 M, Duchefa Biochemie B.V, Haarlem, The Netherlands), BAP (0.8 µM, Sigma-Aldrich, St. Louis, MO, USA), plant agar (0.7%, *w*/*v*, Duchefa) and pH adjusted to 5.7 before autoclaving at 121 °C for 20 min. Plants were subculture monthly and kept in a growth chamber at 25 °C, in a 16 h photoperiod, at 25–35 μmol m^−2^ s^−1^ (white cool fluorescent lamps).

### 4.2. Shoot Regeneration and Culture Conditions

Apical leaves of tamarillo shoots (3 weeks subcultures) cut in approximately 0.25 cm^2^ square segments including the midrib were used for shoot regeneration. MS medium supplemented with sucrose (0.07 M, Duchefa), BAP (8.8 µM, Sigma), plant agar (0.7%, *w*/*v*, Duchefa) and pH adjusted to 5.7 before autoclaving at 121 °C for 20 min was used to induce regeneration. Leaf explants were cultured with the abaxial side down in dark conditions at 24 °C for 3 weeks. After 3 weeks, the cultures were transferred to a 16 h photoperiod at 25–35 μmol m^−2^ s^−1^ and 25 °C for 5 weeks. 

To test the effect of ethylene modulation on shoot regeneration, 10 µM of AgNO_3_ (Merck, Darmstadt, Germany), AVG (Sigma), ACC (Sigma) or ETH (Sigma) were added to the medium. All these modulators were sterilized by filtration with a 0.2 µm filter and added to the medium after autoclaving to avoid thermal degradation. At the end of 8 weeks, regeneration percentage ((number of responsive explants/total number of initial explants) × 100) and the number of shoots and buds developed in responsive explants were analyzed. Three biological replicates were made for the control and each treatment. Each replicate consisted of 8 glass jars fully closed with 3 explants (*N* = 24), in a total of 72 explants per condition. 

### 4.3. Total RNA Isolation and Quantitative PCR Analysis

Samples from responsive explants with 3 and 8 weeks of culture were selected, frozen in liquid N_2_ and stored at −80 °C until RNA extraction. Samples (80 mg) were carefully collected from the visible regeneration sites of the explant. RNA was extracted using the kit NucleoSpin^®^ RNA Plant (MACHEREY-NAGEL GmbH & Co. KG, Duren, Germany) following the manufacturer’s instructions. The final concentration of RNA of each sample was measured using a spectrophotometer (NanoDrop One, Thermo Scientific, MA, USA) and its purity was confirmed with the A_260_/A_280_ and A_260_/A_230_ ratios. RNA integrity was further validated using the Qubit™ RNA IQ Assay Kit (Invitrogen™, Thermo Fisher Scientific, MA, USA). 

First-strand cDNA synthesis was produced from 1 μg of total RNA from 3 biological replicates for each treatment and time-point using the NZY First-Strand cDNA Synthesis Flexible Pack (NZYTech, Lda.—Genes and Enzymes, Lisbon, Portugal) according to the manufacturer’s instructions. Quantitative PCR gene expression analysis of two genes coding for ethylene biosynthetic enzymes, *1-AMINOCYCLOPROPANE-1-CARBOXYLIC ACID SYNTHASE1* (*ACS1*) and *1-AMINOCYCLOPROPANE-1-CARBOXYLIC ACID OXIDASE-HOMOLOG 1* (*ACO1*), the transcription factor *ETHYLENE-RESPONSIVE TRANSCRIPTION FACTOR* 61 (*ERF061*) and the auxin efflux carrier *PIN-FORMED1* (*PIN1*), was made using NZYSpeedy qPCR Green Master Mix (2×) (NZYTech, Lda.—Genes and Enzymes, Lisbon, Portugal), following the instructions provided with 50-fold diluted cDNA template. Reactions were performed in a 96-well plate, with two technical replicates measured in CFX96 Real-Time System (Bio-Rad, CA, USA). Gene expression was normalized for both *IRON SUPEROXIDE DISMUTASE, FeSOD* and ACTIN, *ACT* reference genes [79]. All the primers (Table 2), except for *PIN1* gene primers, were designed from *Solanum betaceum* transcript sequences obtained from embryogenic cell RNA-seq libraries (unpublished data), using the NCBI primer design tool. *PIN1* gene primers were designed from the reference sequence of *Solanum lycopersicum* (NM_001247234.2) after the selection of conserved coding regions based on the alignment of *Solanum* sp. sequences (*S. lycopersicum*; *S. pennellii,* XM_015212230.2 and *S. tuberosum,* XM_006341465.2). The relative expression was calculated according to the Pfaffl method [80], using non-treated explants as a control for each timepoint.

### 4.4. Rooting and Acclimatization

Tamarillo shoots regenerated from leaf explants with at least 1 cm were in vitro rooted in hormone-free MS medium supplemented with sucrose (0.07 M, Duchefa), plant agar (0.7%, *w*/*v*, Duchefa) and pH adjusted to 5.7 before autoclaving at 121 °C for 20 min. After 1 month, rooting percentage ((number of shoots with roots/number of initial shoots) × 100) for each treatment was analyzed.

Plants were acclimatized in a walk-in chamber (FitoClima 10000 HP, Aralab) under 16 h photoperiod at 40 μmol m^−2^ s^−1^, 25 °C and 70% humidity. Briefly, plant roots were carefully washed to remove agar debris and placed on covered containers (70 cm³) with Siro Royal substrate (SIRO, Mira, Portugal). The cover was removed after 2 weeks, and after 1 month, the plants were transferred to individual containers (500 cm^3^) and the acclimatization percentage ((number of survival plants/number of initial plants) × 100) was analyzed. Plant height was evaluated after 1- and 3-months ex vitro. At the end of 3 months, dry matter percentage ((dry weight/fresh weight) × 100) was also assessed. For this purpose, 3-months acclimatized plants were carefully washed, weighted, and dried for 48 h at 70 °C.

### 4.5. Statistical Analysis

All data are presented as mean ± SEM and statistical analysis was performed using GraphPad Prism 9. The differences between treatments were analyzed using one-way ANOVA, followed by a Tukey’s multiple comparison test. For gene expression analysis, differences of each treatment relative to the control were analyzed by the Student’s *t* test.

## 5. Conclusions

In conclusion, our findings indicate that ethylene plays a crucial role in DNSO in ta-marillo. When ethylene action is inhibited, both the formation of organogenic callus and the regeneration of shoot-buds are reduced. These results suggest the possibility of a cyto-kinin-ethylene-auxin crosstalk that promotes callus formation and subsequent shoot re-generation. The upregulation of *ERF061* suggests that ethylene can affect shoot regeneration through stress-response signaling. Additionally, the upregulation of *PIN1* by ethylene supports previous reports and implies that ethylene may enhance shoot regeneration by affecting auxin distribution prior to shoot development. To enhance our comprehension of the molecular mechanisms driving the impact of ethylene on regeneration, it would be valuable to investigate the distribution and quantification of auxins, alongside other regeneration-related genes. Overall, this first approach of the ethylene effect on in vitro regeneration of tamarillo sheds light on the possible molecular mechanisms involved in the induced shoot regeneration of woody species.

## Figures and Tables

**Figure 1 plants-12-01854-f001:**
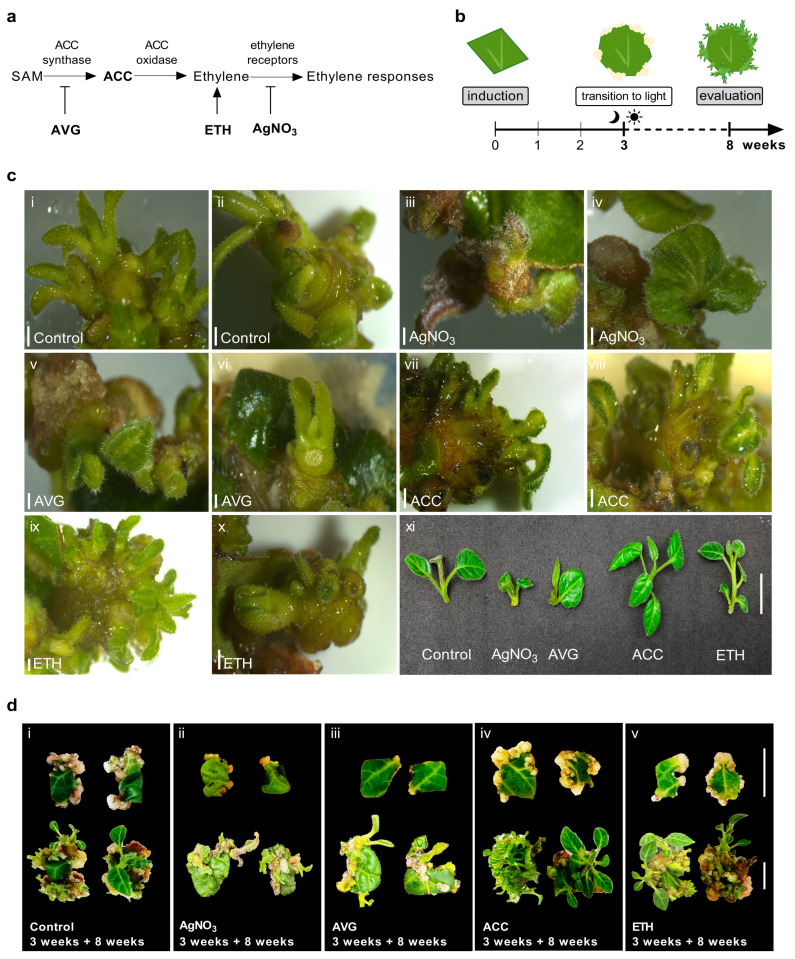
Ethylene availability enhances shoot-bud regeneration from leaf explants in tamarillo. (**a**) Schematic representation of the compounds used in this study and their role in ethylene biosynthesis, signaling and perception. (**b**) A schematic overview of the experimental set-up for shoot regeneration. Regeneration was induced from leaf explants in the dark, in the presence of 10 µM of each different modulator (AgNO_3_, AVG, ACC and ETH). After 3 weeks of culture, the explants were transferred to a 16 h photoperiod. At the end of the 8th week of culture, regeneration percentage and the number of buds and shoots induced were analyzed. (**c**) Bud and shoot regeneration after 8 weeks of culture in the different treatments. *i*–*x*: scale bars, 1 mm. *xi*: scale bar, 1 cm. (**d**) Explants after 3 (top) and 8 (bottom) weeks of culture. Scale bars, 1 cm.

**Figure 2 plants-12-01854-f002:**
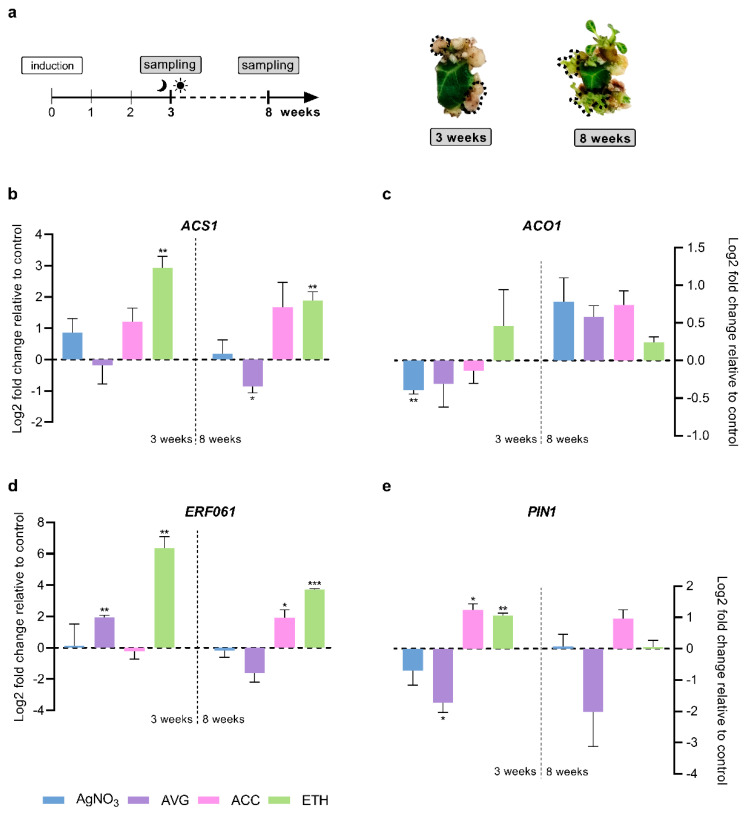
Effect of the different modulators on gene expression of *ACS1*, *ACO1*, *ERF061* and *PIN1* at two selected culture timepoints. (**a**) A schematic overview of the experimental set-up for the analysis of gene expression. Samples from control, AgNO_3_, AVG, ACC and ETH conditions were collected at the 3rd and 8th week of culture. Only explant sites with regeneration responses were collected, as shown by the dashed lines. (**b**–**e**) Fold-relative gene expression compared to each control condition (represented as zero with dashed line). Data are represented as mean ± SEM (n = 3). Asterisks indicate statistically significant differences between treatments and control (Student’s *t* test, * *p* < 0.05, ** *p* < 0.01 and *** *p* < 0.001).

**Figure 3 plants-12-01854-f003:**
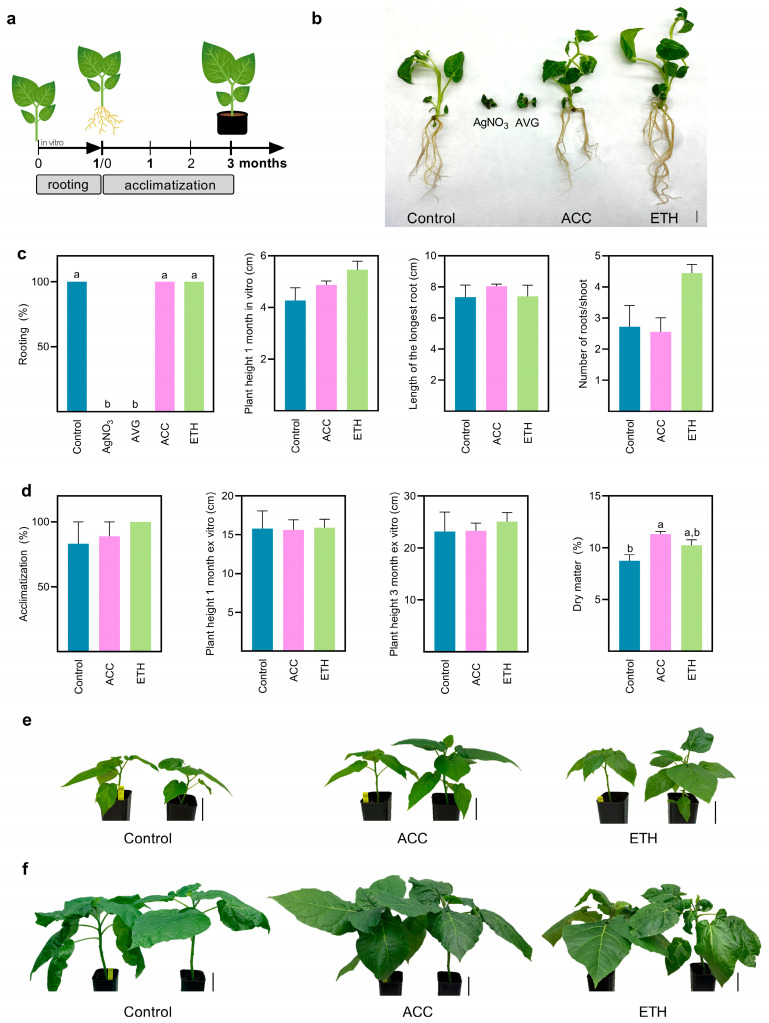
Rooting and acclimatization of regenerated shoots induced in each treatment. (**a**) A schematic overview of the experimental set-up for rooting and acclimatization assays. Shoots were in vitro rooted in hormone-free MS medium (sucrose, 3%, *w*/*v*) without modulators. After 1 month of culture, shoots were analyzed and rooted plants were acclimatized. (**b**) Shoots regenerated from each condition after 1 month in hormone-free MS medium. Scale bars, 1 cm. (**c**) Rooting percentage and morphological parameters after 1 month in hormone-free MS medium. (**d**) Acclimatization percentage and physiological parameters after 1- and 3-months ex vitro. (**e**) Acclimatized plants regenerated from control, ACC and ETH conditions after 1 month ex vitro. Scale bars, 5 cm. (**f**) Acclimatized plants after 3 months ex vitro. Scale bars, 5 cm. For rooting rate, in vitro parameters and acclimatization rate, data are represented as mean ± SEM of three biological replicates (n = 3, N = 6 shoots or plants per replicate). For ex vitro parameters, data are represented as mean ± SEM (n = 6 plants per condition). Letters indicate statistically significant differences between treatments (one-way ANOVA with Tukey multiple comparison test, *p* < 0.05).

**Table 1 plants-12-01854-t001:** Effect of ethylene modulators on regeneration percentage and number of buds and shoots per explant after 8 weeks.

Treatment	Regeneration Percentage (%)	No of Buds and Shoots per Explant	Observations
Control	73.61 ± 6.05 ^a^	12.21 ± 1.19 ^b^	Presence of well-developed and elongated shoots with fully opened leaves.
AgNO_3_	65.28 ± 6.05 ^a^	6.36 ± 1.18 ^c^	Shoots not completely developed nor elongated. Some leaves were fully open but presented abnormal shape.
AVG	48.61 ± 1.39 ^a^	6.40 ± 1.16 ^c^	Shoots neither developed nor elongated. Some leaves were fully opened.
ACC	66.67 ± 6.37 ^a^	19.09 ± 1.03 ^a^	Presence of well-developed and elongated shoots with fully opened leaves.
ETH	58.33 ± 6.37 ^a^	18.00 ± 1.29 ^a^	Presence of well-developed and elongated shoots with fully opened leaves.

For regeneration percentage data are represented as mean ± SEM of three biological replicates (n = 3, N = 24 explants per replicate). For number of buds and shoots data are represented as mean ± SEM of three replicates (n = 3, 15 < N < 20 explants analyzed per replicate for control, 13 < N < 18 for AgNO_3_, 11 < N < 12 for AVG, 13 < N < 18 for ACC, 12 < N < 17 for ETH). Letters indicate statistically significant differences between treatments (one-way ANOVA with Tukey multiple comparison test, *p* < 0.05).

**Table 2 plants-12-01854-t002:** Primer pairs used for gene expression analysis.

Gene	Forward Primer	Reverse Primer
*ACO1*	GCTAACTCTTGGAGCTGGCA	GCCACTACTCTGTGTGCAGT
*ACS*	TCCACAGTGAATCCCATTTTGAT	GGCTTAGCTTTGTTCTTTGTTGT
*ACT*	CCATGTTCCCGGGTATTGCT	GTGCTGAGGGAAGCCAAGAT
*ERF061*	TCTTCGCGATCCAAGCAAGT	ACCACCACCAACCAAAGAAGA
*FeSOD*	TCACCATCGACGTTTGGGAG	GACTGCTTCCCATGACACCA
*PIN1*	ACCAAGGATCATAGCATGTGGA	CTTGTGGTAGAGCTGCCTGT

## Data Availability

The original contributions presented in the study are included in the article. Further inquiries can be directed to the corresponding author.

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
