# Peer review of "Ethylene Inhibition Reduces De Novo Shoot Organogenesis and Subsequent Plant Development from Leaf Explants of *Solanum betaceum* Cav."

_plants, 2023, doi:10.3390/plants12091854_

Round 1

Reviewer 1 Report

I have not particular comments to the authors. I strongly suggest the publication of this manuscript.

On my 

Author Response

We wanted to express our sincerest gratitude for your time and effort in reviewing our scientific article. Your positive feedback and recommendation for the article to be published was truly appreciated.

Best regards.

Reviewer 2 Report

In the Introduction section, the authors make a controversial statement about the acquisition of the property of pluripotency by cells when auxin is added. However, there is no rigorous evidence from which cells (and further, groups of cells) shoots develop. There is an alternative point of view that regeneration can only come from stem cells. Stem cells (at least having the properties of stem cells) could be companion cells adjacent to the vascular bundle in the cut plant explant. It is advisable to discuss this point of view in at least a few sentences.

The conclusion contains a provision on the effect of ethylene on the distribution of auxin. However, the authors of the study did not define this in their work. Therefore, this provision needs to be mitigated.

In general, the work was done at a good level and deserves to be accepted into the journal Plants with minor revisions.

Author Response

We would like to extend our appreciation for your thorough review of our scientific article. Your insightful comments and constructive feedback have been invaluable in improving the quality of our work.

Point 1: In the Introduction section, the authors make a controversial statement about the acquisition of the property of pluripotency by cells when auxin is added. However, there is no rigorous evidence from which cells (and further, groups of cells) shoots develop. There is an alternative point of view that regeneration can only come from stem cells. Stem cells (at least having the properties of stem cells) could be companion cells adjacent to the vascular bundle in the cut plant explant. It is advisable to discuss this point of view in at least a few sentences.

Response 1: We add few sentences regarding the type and group of cells callus and shoots developed:
"There is compelling evidence that callus formation and the subsequent ability to regenerate is derived from specific cell types that act as potential pluripotent stem cells. Auxin-induced callus appears to originate from pericycle and pericycle-like cells located around the vasculature, while wound-induced callus can arise from various cell types such as epidermis, xylem parenchyma, procambium, and mesophyll. In some species, de novo shoots and roots regenerate from procambium or cambium cells."

Point 2: The conclusion contains a provision on the effect of ethylene on the distribution of auxin. However, the authors of the study did not define this in their work. Therefore, this provision needs to be mitigated.

Response 2: We adjusted the sentence from ". Additionally, the upregulation of PIN1 by ethylene supports previous reports and implies that ethylene enhances shoot regeneration by affecting auxin distribution prior to shoot development" to ". Additionally, the upregulation of PIN1 by ethylene supports previous reports and implies that ethylene may enhances shoot regeneration by affecting auxin distribution prior to shoot development" and suggested investigating this aspect in future studies in the following sentence: "To enhance our comprehension of the molecular mechanisms driving the impact of ethylene on regeneration, it would be valuable to investigate the distribution and quantification of auxins, alongside other regeneration-related genes.".

Your positive feedback was truly appreciated.

Best regards. 

Reviewer 3 Report

Dear Authors, 

I have read your manuscript with great interests. I think that the study of the molecular mechamisms involved in de novo shoot organogenesis is a crucial step to understand these processes, and to transfer the knowledge to other recalcitrant species. 

Attached you will find the revised manuscript, with just a comment. 

As far I am concerned, the manuscript is ready to be published. 

Best regards 

Author Response

We would like to extend our appreciation for your thorough review of our scientific article. 

Point: The sentence in the abstract “Additionally, it was found that shoots regenerated where ethylene biosynthesis and perception was inhibited failed to develop.” was not clear.

Response: We rewrite the sentence to “Additionally, it was found that the de novo shoot meristems induced in explants in which ethylene biosynthesis and perception was suppressed were unable to further develop into elongated shoots.” Hopefully the message is clearer now.

Your positive feedback was truly appreciated.

Best regards.